# Effects of Diets Based on Hydrolyzed Chicken Liver and Different Protein Concentrations on the Formation and Deamination of Biogenic Amines and Total Antioxidant Capacity of Dogs

**DOI:** 10.3390/ani13162578

**Published:** 2023-08-10

**Authors:** Caroline Fredrich Dourado Pinto, Camila Figueiredo Carneiro Monteiro, Marcelino Bortolo, Fábio Ritter Marx, Jorge Felipe Argenta Model, Anapaula Sommer Vinagre, Luciano Trevizan

**Affiliations:** 1Department of Animal Science, Universidade Federal do Rio Grande do Sul, Porto Alegre 91540-000, Brazil; krolfredrich@hotmail.com (C.F.D.P.); camilafcm@hotmail.com (C.F.C.M.); 2Nutrisurance Division, Kemin Industries, Inc., Indaiatuba 13347-394, Brazil; marcelino.bortolo@kemin.com; 3Nutrisurance Division, Kemin Industries, Inc., Des Moines, IA 50317, USA; fabio.marx@kemin.com; 4Comparative Metabolism and Endocrinology Laboratory (LAMEC), Department of Physiology, Universidade Federal do Rio Grande do Sul, Porto Alegre 90035-003, Brazil; jorgefamodel@gmail.com (J.F.A.M.); anapaula.vinagre@ufrgs.br (A.S.V.)

**Keywords:** biogenic amines, diamine oxidase, dog diets, hydrolyzed chicken liver powder, monoamine oxidase, poultry by-product meal

## Abstract

**Simple Summary:**

Biogenic amines (BAs) originate from amino acid decarboxylation and are commonly used as markers of spoilage and putrefaction in food and feed, especially in improperly handled animal products. Detoxification of BAs occurs mainly in the intestinal tract by amine oxidases, but some factors may facilitate intestinal absorption. When consumed in high concentrations, BAs can impair colonic epithelium, intestinal functionality, and may cause other toxicological effects. This study aimed to evaluate the effects of diets based on hydrolyzed protein (hydrolyzed chicken liver powder—HCLP) or conventional intact protein (poultry by-product meal—PBPM) with increasing protein concentrations (24, 32, and 40%) on the consumption and metabolism of biogenic amines in adult dogs. The results showed that PBPM had higher concentrations of putrefactive BAs, as well as the diets with its inclusion. As a result, dogs consumed higher concentrations of putrefactive BAs with diets containing PBPM and higher protein concentrations. Despite the difference in BAs concentrations in the protein sources, there was only an increase in the fecal excretion of phenylethylamine in the HCLP32 and HCLP40 diets and the enzymatic activity of monoamine oxidase in the HCLP24 and PBPM32 diets. Our findings indicate that the inclusion of HCLP reduces the consumption of putrefactive BAs and that the BAs detoxification system is highly efficient in the intestinal tract of dogs.

**Abstract:**

Biogenic amines are synthesized through the bacterial decarboxylation of amino acids, commonly found in high levels in animal by-product meals due to spoilage. Furthermore, biogenic amines and other metabolites can be produced by the fermentation of proteins in the hindgut according to the protein source and concentration of crude protein (CP) in the diet. Thus, this study aimed to evaluate two protein sources (poultry by-product meal (PBPM) and hydrolyzed chicken liver powder (HCLP)) and three CP concentrations (24, 32, and 40%) and their effects on the consumption and fecal excretion of biogenic amines, plasma monoamine oxidase (MAO) and diamine oxidase (DAO) activities, and total antioxidant capacity (TAC) of healthy adult dogs after 30 days of feeding the experimental diets. Twelve dogs were randomly distributed into six treatments (n = 6/treatment): PBPM24 (PBPM with 24% CP); PBPM32 (PBPM with 32% CP); PBPM40 (PBPM with 40% CP); HCLP24 (HCLP with 24% CP); HCLP32 (HCLP with 32% CP); HCLP40 (HCLP with 40% CP). The PBPM and PBPM-based diets had higher concentrations of putrescine, cadaverine, tyramine, histamine, agmatine, and total biogenic amines. In contrast, HCLP and HCLP-based diets contained higher concentrations of spermidine, phenylethylamine, and spermine. The PBPM and PBPM-diets had higher biogenic amine index (BAI) indicating lower quality due to the high content of putrescine, cadaverine and tyramine. Dogs fed diets with PBPM and higher protein concentrations consumed more putrescine, cadaverine, tyramine, agmatine, and total amines (*p* < 0.0001), while dogs fed with HCLP consumed more spermidine, phenylethylamine, and spermine (*p* < 0.0001). Fecal excretion of phenylethylamine was greater in dogs fed HCLP32 and HCLP40 diets (*p* = 0.045). Dogs fed with HCLP tended to excrete more spermidine and tryptamine via feces, while higher protein concentrations tended to increase fecal excretion of cadaverine (*p* < 0.10). Plasma MAO activity was higher in dogs fed HCLP24 and PBPM32 diets (*p* = 0.024). The plasma activities of DAO and TAC were not different between diets (*p* > 0.05). Although we did not evaluate the intestinal activities of MAO and DAO, our results suggest that healthy adult dogs have an efficient deamination process on the gut epithelium.

## 1. Introduction

Biogenic amines are non-volatile low molecular weight nitrogenous organic bases, originated from decarboxylation of amino acids. In humans and animals, biogenic amines act on the regulation of body temperature, stomach volume, stomach pH and neuronal activity [1]. However, consumption of high concentrations of certain biogenic amines, particularly histamine and tyramine, can cause toxic effects [2]. Biogenic amines formation requires the availability of free amino acids, the presence of decarboxylase-positive microorganisms, time and temperature during storage and processing [1,3]. In this way, biogenic amines have been widely used as food and feed quality markers to indicate their degree of freshness and/or spoilage [2,3].

Animal by-products are commonly used in companion animal diets, especially poultry and meat by-products, which may have undergone some degree of spoilage and therefore could be considered a high source of biogenic amines [2]. Thus, poultry by-product meal (PBPM) diets can increase the consumption of biogenic amines in dogs. On the other hand, the use of hydrolyzed proteins has been increasing due to their high concentrations of small peptides and amino acids, easier digestion and absorption, and for containing bioactive peptides with functional properties such as antimicrobial, antioxidant, antihypertensive and immunomodulatory activities [4,5,6]. Thus, considering that the production of hydrolyzed proteins involves the use of highly technological methods to improve final product quality [6], we hypothesized that diets based on poultry by-product meal could increase the consumption of biogenic amines, mainly amines associated with the deterioration process of food and feed.

To the authors knowledge, there are no studies associating the levels of biogenic amines in protein sources and diets and their effects on food intake, detoxification, and fecal excretion in dogs. Therefore, the present study investigated the concentrations of biogenic amines in different protein sources (PBPM and hydrolyzed chicken liver powder (HCLP)) after their inclusion in experimental diets containing increasing dietary protein concentrations (24, 32, and 40% of crude protein). Further, we also evaluated the effects of consumption and fecal excretion of biogenic amines, plasma monoamine oxidase (MAO) and diamine oxidase (DAO) activities and total antioxidant capacity (TAC) in healthy adult dogs after 30 days of feeding with these experimental diets.

## 2. Materials and Methods

### 2.1. Animals and Housing

Twelve healthy, intact adult Beagle dogs (six males and six females), with 2.95 ± 0.88 years of age, weighing 11.2 ± 0.85 kg, with a body condition score ranging from 4.5 to 6.5 out of 9 points [7] were used in a balanced incomplete Latin square design [8] with three 30 d blocks and two dogs per diet in each block (n = 6/treatment). Dogs were submitted to clinical and physical examination, de-wormed and vaccinated before the trial beginning. All dogs were housed individually into stainless steel metabolic cages (1.0 × 1.0 × 1.5 m) equipped with a feces and urine collector, feeders, and drinkers, in a controlled room at 24 °C, with a light: dark cycle of 14:10 h. During the study, dogs were fed twice daily inside the metabolic cages and stayed there throughout the night. During the day, the dogs remained all together in an outdoor area for socialization.

### 2.2. Diets

Dogs were fed a commercial diet for 30 days prior the study beginning (wash-out period), to stabilize the gastrointestinal tract. Six dry, extruded (TNL Tecnal, model 2000, Ourinhos, Brazil), and complete diets were formulated to exceed the minimum nutrient requirements for adult dogs as recommended by the European Pet Food Industry Federation Nutritional Guidelines (FEDIAF) [9]. Diets were manufactured at Nutribarrasul Alimentos Pet LTDA (Barra do Ribeiro, Rio Grande do Sul, Brazil). Diets varied in protein source (poultry by-product meal (PBPM—purchased from a rendering plant in the South of Brazil) and hydrolyzed chicken liver (HCLP—PROSURANCE CHX Liver HD, Kemin Industries, Vargeão, Santa Catarina, Brazil)) and concentration (24, 32 and 40% of protein) (Table 1). Diets were: (1) PBPM24—diet based on PBPM with 24% of protein; (2) PBPM32—PBPM diet with 32% of protein; (3) PBPM40—PBPM diet with 40% of protein; (4) HCLP24—diet based on HCLP with 24% of protein; (5) HCLP32—HCLP diet with 32% of protein; (6) HCLP40—HCLP diet with 40% of protein. The metabolizable energy (ME) of the diets was estimated using the equation proposed by the National Research Council (NRC) [10]: ME (kcal/kg) = (4 × g protein) + (9 × g fat) + (4 × N-free extract) × 1000. Dogs were fed twice a day (at 08:30 h and 16:00 h) to meet their energetic and nutritional requirements, as recommended by the NRC [10]. Food intake was weekly adjusted according to body weight to maintain the body condition score in 5 points out of 9 [7]. Water was provided ad libitum throughout the study.

### 2.3. Biogenic Amine Analysis

Protein sources (PBPM and HCLP), diets and feces were analyzed for biogenic amine concentrations based on the methodology by Vale and Gloria [11]. The concentrations of biogenic amines in the feces were evaluated after 30 days of feeding with the diets in each block (days 30, 60 and 90, respectively). Standards for biogenic amines (putrescine dihydrochloride, cadaverine dihydrochloride, tyramine hydrochloride, histamine dihydrochloride, serotonin hydrochloride, agmatine sulfate, spermidine trihydrochloride, 2-phenylethylamine hydrochloride, spermine tetrahydrochloride, tryptamine) were from Sigma-Aldrich Chemical Company (St. Louis, MO, USA). Analytical grade sodium acetate trihydrate, glacial acetic acid, octane sulfonic acid sodium salt, boric acid, Brij-35 (30% *w*/*v*), β-mercaptoethanol, potassium hydroxide and *o*-phthalaldehyde (OPA) were from Sigma-Aldrich. Acetonitrile, methanol, and hexane were liquid chromatography grade. Ultrapure water was from Milli-QTM (Millipore Corporation, Milford, MA, USA). Organic and aqueous solvents for HPLC analysis were filtered through 0.45 μm pore size HVPL Membranes (Millipore Corporation, Milford, MA, USA). Five grams of each sample were homogenized and added to 7 mL of 5% trichloroacetic acid (TCA, 5% *w*/*v*), mixed at 280 rpm for 10 min by vortex and centrifuged at 10,000× *g* for 20 min at 4 °C (MR23I Jouan, refrigerated centrifuge, Saint Herblain, France). The supernatant was filtered with qualitative filter paper, collected into a 25 mL volumetric flask and then completed with 5% TCA. After homogenization, an aliquot of the supernatant was filtered using a syringe, swinnex filter holder, and cellulose ester membrane immediately prior to high performance liquid chromatography (HPLC) analysis. Biogenic amine contents were determined in the supernatant by HPLC (Shimadzu, model LC-20AD, Kyoto, Japan), coupled to a fluorescence (Shimadzu RF-10AXL) detector at 340 nm excitation and 450 nm emission with a CBM-20A HPLC interface control unit and a SIL-20AHT auto-injector. Chromatographic separations were obtained in the Luna C18 Phenomenex column (4.6 × 250 mm, 5 μm) and a C18 pre-column (4 × 3 mm) in an oven (CTO-10 ASvp, Shimadzu) at 30 °C. A gradient elution of (A) 0.2 M sodium acetate and 15 mM octane sulfonic acid sodium salt, pH adjusted to 4.9 with acetic acid, and (B) acetonitrile was used at: 0.01–17.99 min/2% B; 18.00–18.99 min/20% B; 19.00–39.99 min/5% B; 40.00–49.99 min/23% B; 50.00–50.49 min/35% B; 50.50–60.00 min/2% B. The post column derivation reagent, delivered at 0.3 mL/min, consisted of 1.5 mL Brij-35, 1.5 mL β-mercaptoethanol and 0.2 g OPA dissolved in 500 mL solution of 25 g boric acid and 22 g KOH (pH adjusted to 10.5 with 3% KOH). Total analysis time was 60 min. Identification of biogenic amines was performed by comparison of the retention time of the analyte peaks in the sample with those of the standard solution and by addition of the suspected biogenic amine to the sample. Quantification of biogenic amines was calculated by interpolation in the respective external analytical curves.

### 2.4. Plasma Monoamine Oxidase (MAO) and Diamine Oxidase (DAO) Activities and Total Antioxidant Capacity (TAC)

Blood samples were collected after 30 days of feeding with the diets in each block (days 30, 60 and 90, respectively), at the morning before food consumption. Dogs were individually placed over a table and the blood samples were collected via cephalic vein puncture and placed on tubes with EDTA (MAO and DAO) and heparin (TAC), a total of 2 tubes of 3 mL each were collected per dog on each day of collection. The tubes were centrifuged at 3000× *g* during 15 min within 30 min of blood collection, then the plasma was transferred to Eppendorf tubes and stored frozen at −20 °C until MAO, DAO, and TAC analyses. The plasma MAO and DAO activities were performed using fluorometric kits (Monoamine Oxidase Assay Kit ab241031, Abcam^®^ and Diamine Oxidase Assay Kit ab241004, Abcam^®^, Cambridge, UK, respectively) according to the manufacturer’s instructions. Total antioxidant capacity (TAC) was quantified using colorimetric kit according to the manufacturer’s instructions (QuantiChrom Antioxidant Assay Kit DTAC-100, BioAssay Systems, Hayward, CA, USA).

### 2.5. Calculations and Statistical Analysis

Based on the laboratory results, the biogenic amine index (BAI) of protein sources and diets were calculated by summing putrescine, cadaverine, tyramine, and histamine concentrations, and classified as good quality (BAI < 5 mg/kg), acceptable (5 mg/kg < BAI < 20 mg/kg), low quality (20 mg/kg < BAI < 50 mg/kg) and spoiled (BAI > 50 mg/kg) [3]. Consumption of biogenic amines was calculated as: Biogenic amine intake (mg/kg as is) = biogenic amine content in diet (mg/kg) × feed intake (g)/1000. Furthermore, the balance of biogenic amines was calculated based on the difference between the amount of biogenic amine consumed and excreted in feces.

Data were tested for homogeneity of variances and normality of errors. When necessary, natural logarithmic transformation (ln) was applied to achieve normality. Then data were analyzed using two-way ANOVA complemented by Tukey’s test to compare the protein source (PBPM or HCLP), concentration (24, 32 or 40% CP), and the interaction (protein source and concentration). The analysis also included block as a fixed effect and dog as a random effect in Minitab 18 (Minitab Inc., State College, PA, USA). A probability of *p* < 0.05 was accepted as statistically significant, while *p* < 0.10 was considered a tendency.

## 3. Results

### 3.1. Biogenic Amines in Protein Sources and Diets

Dogs consumed their entire daily ration without refusals and remained healthy throughout the study without episodes of vomiting and diarrhea. The biogenic amines concentrations differed between protein sources (PBPM and HCLP) and diets (Table 2). The PBPM had the highest concentrations of putrescine, cadaverine, tyramine, histamine, agmatine, and total biogenic amines. Conversely, HCLP had the greatest concentrations of spermidine, phenylethylamine, and spermine. There was an increase in the content of biogenic amines in all diets due to increasing dietary protein concentrations by increasing the inclusion of PBPM and HCLP. Regarding the quality, both protein sources showed high BAI (BAI > 50 mg/kg) and were classified as spoiled. Based on the BAI, only the HCLP24 diet was considered of good quality (BAI < 5 mg/kg).

### 3.2. Biogenic Amines Intake and Fecal Excretion

Consumption of biogenic amines, calculated by the content of biogenic amines in each diet and feed intake, differed between treatments (Table 3). Dogs fed PBPM consumed higher concentrations of putrescine, cadaverine, tyramine, histamine, agmatine, and the sum of total amines than those fed HCLP-based diets (*p* < 0.0001). In the diets containing higher protein concentration, regardless of the type of diet, the consumption of the analyzed amines, with exception of spermidine, and the sum of total amines (*p* < 0.0001) were also higher (Table 3). When the type of protein source and the concentration of protein were analyzed together, i.e., interaction effects, dogs fed with PBPM at 32 and 40% CP consumed more putrescine, cadaverine, tyramine, histamine, agmatine, and the sum of total amines than dogs fed HCLP-based diets (*p* < 0.0001). Conversely, dogs fed HCLP with higher protein concentrations consumed more spermidine, phenylethylamine, and spermine (*p* < 0.0001).

Fecal excretion of spermidine and tryptamine tended to increase in dogs fed with HCLP (*p* < 0.10) (Table 4). Also, fecal excretion of cadaverine tended to increase in dogs fed with higher protein concentrations (*p* = 0.080). Lastly, dogs fed with HCLP at 32 and 40% CP had greater fecal excretion of phenylethylamine (*p* = 0.045).

### 3.3. Balance of Biogenic Amines

The balance of biogenic amines, calculated as the differences between the concentration of biogenic amines consumed and excreted via feces, revealed effects of protein source, concentration, and the interaction (Table 5). In this way, we consider that a positive balance represents an increased production of the biogenic amine, while a negative balance represents that the biogenic amine was detoxified in the intestinal tract.

Dogs fed diets with PBPM had the greatest putrescine, cadaverine, tyramine, histamine, agmatine, and the sum of total amines balance (*p* < 0.0001). Also, dogs fed with PBPM tended to have a greater tryptamine balance (*p* = 0.050). While spermidine, phenylethylamine, and spermine balance was greater in dogs fed with HCLP (*p* < 0.0001).

The balance of putrescine, cadaverine, tyramine, histamine, agmatine, spermidine, phenylethylamine, spermine, and the sum of total amines increased with higher protein concentrations (*p* < 0.0001) (Table 5). There were interaction effects between protein source and concentration, observed as increases in the balance of putrescine, cadaverine, tyramine, histamine, agmatine, and the sum of total amines in dogs fed with PBPM at 32 and 40% CP (*p* < 0.0001). In addition, interaction effects were observed as increases in the balance of spermidine, phenylethylamine and spermine in dogs fed with HCLP at 32 and 40% CP (*p* < 0.0001).

### 3.4. Plasmatic Activities of MAO, DAO, and TAC

Plasmatic activity of MAO was higher in dogs fed HCLP at 24% CP and PBPM at 32% CP (*p* = 0.024) (Figure 1A). Plasmatic concentrations of DAO and TAC were not affected by protein source, concentration, and their interaction (*p* > 0.05) (Figure 1B,C).

## 4. Discussion

Biogenic amines have been widely used as quality indexes in food and feed, given their close relationship with nutritional, health and safety aspects [2,3,12,13]. Although biogenic amines participate in several physiological processes, consumption of high concentrations of certain biogenic amines, mainly histamine and tyramine, can result in a variety of toxicological effects and health damage [14]. Animal by-products are traditionally included in pet food due to their high nutritional value and acceptability, but they are potential sources of biogenic amines if they undergo any degree of deterioration [2,13]. Currently, there is a lack of studies that evaluate the effects of biogenic amines in animal by-products on the metabolism and formation of these metabolites in dogs despite its importance for feed safety. Therefore, the present study investigated the effects of consuming diets containing protein sources with varying concentrations of biogenic amines on detoxifying enzyme activity and fecal excretion in healthy adult dogs.

Our results revealed that the PBPM had the greatest concentrations of biogenic amines associated with spoilage and putrefaction. Thus, dogs fed diets with PBPM and increasing dietary protein consumed more putrescine, cadaverine, tyramine and histamine compared to those fed with HCLP. These findings agree with Tamim and Doerr [15], in which it was demonstrated that poultry carcasses submitted to progressive putrefaction (up to 72 h) until rendering, had high concentrations of putrescine, cadaverine, histamine, tyramine, tryptamine and phenylethylamine. In accordance, a recent study showed that poultry meals from four different rendering plants had the greatest diversity of biogenic amines compared to other animal meals [13]. Hence, it is recommended that animal by-products should be rendered immediately after slaughter, especially in locations with high temperatures, to avoid the production of high concentrations of biogenic amines [2]. According to the BAI classification, only HCLP24 diet (1 mg/kg) was considered of good quality while the other diets were considered spoiled/deteriorated (BAI > 50 mg/kg). However, as mentioned by Ruiz-Capillas and Herrero [3], defining a BAI that reliably predicts product quality depends mainly on the nature of the product (e. g. fresh, heat-treated). In this way, the use of free biogenic amines seems to provide a better understanding of the quality of animal by-products.

High concentrations of amino acid fermentation metabolites, including biogenic amines, may exert negative effects on the colonic epithelium and intestinal functionality [16]. Several factors influence the synthesis and final concentration of biogenic amines, such as the characteristics of the raw material (composition, mainly the availability of free amino acids, and pH), the presence of microorganisms with decarboxylase activity (*Enterobacteriaceae*, *Pseudomonadaceae*, *Micrococcaceae*, lactic acid bacteria, etc.), hygienic conditions, and processing and storage conditions [1,3]. Our initial concern was that HCLP-based diets would have higher concentrations of biogenic amines due to the high availability of free amino acids generated from the hydrolysis process. However, dogs fed HCLP-based diets with higher protein concentrations (HCLP40 and HCLP32), only consumed more spermidine, phenylethylamine, and spermine than dogs fed PBPM-based diets. Polyamines, spermidine and spermine, participate in cell growth and proliferation, with a potential important role in wound healing and in the development of the digestive system in neonates [17,18]. Interestingly, spermidine and spermine originate mainly from raw materials, while the other biogenic amines can also be formed by protein degradation driven by bacteria. A recent study showed a negative correlation between microbiological counts and spermine concentration during the storage of chicken meat, suggesting that polyamines can be used as a nitrogen source by decarboxylase-positive microorganisms for the synthesis of biogenic amines [19]. Thus, polyamines can be considered freshness indicators, as their concentrations decrease with the deterioration process. Polyamines in the intestinal lumen come from diet, endogenous secretion, synthesis by the intestinal microbiota, and release from colonocyte desquamation. Rats fed a polyamine-deficient diet for an extended period showed colonic mucosal hypoplasia [20], revealing that luminal polyamines in the diet are important factors for colonic mucosal renewal. In this way, the inclusion of HCLP can improve gut development and health in young life stages such as in puppies.

Although no tryptamine was detected in the protein sources and diets, dogs fed the HCLP-based diets tended to have greater fecal excretion of tryptamine compared to those fed with PBPM. Thus, the tryptophan content in the protein source must be considered, as it is the precursor amino acid of tryptamine. In our previous study, the amino acid score (AAS) revealed that the same HCLP ingredient had higher concentrations of tryptophan than a traditional PBPM [21]. Once in intestinal lumen, tryptophan can be absorbed by enterochromaffin (EC) cells and be converted to serotonin (5-hydroxytriptamine, 5-HT), and this is the main site of 5-HT synthesis (90%) in mammalian body [22,23]. These cells can convert 5-HT to 5-hydroxyindoleacetic acid (5-HIAA) by MAO or secrete 5-HT to both intestinal lumen and intercellular space, and then 5-HT can be transported to other intestinal cells, neurons, and blood capillaries [23]. In gut microbes, tryptophan can also be processed by the indole pathway, leading to tryptamine and indole production [22]. Tryptamine may act as a signaling molecule affecting gastrointestinal motility and intestinal transit time [24] but was also linked to the development of insulin resistance in type 2 diabetes [25].

Biogenic amines are efficiently inactivated in the intestinal tract of dogs. According to Kim et al. [26], DAO and imidazole-N-methyl transferase (IMT) activities were high in the mucosa of different segments of the gastrointestinal tract of dogs, from the stomach to the final segment of the large intestine. This may be related to the ancestral feeding behavior of dogs, which were frequently exposed to putrefied meat. Histamine, putrescine and several amines and amino acids such as hydroxyproline can be reduced by DAO, also called histaminase [27]. The absence of difference in DAO activity can be related to the high concentration of amines and amino acids in both types of diets. MAO activity was influenced both by the type of diet and the concentration of proteins and was higher in HCLP 24% and PBPM 32% groups. Both enzymes also produce ammonia, from the deamination process, and hydrogen peroxide (H_2_O_2_), which could have led to oxidative stress [22,23,28]. Therefore, the absence of difference in total antioxidant capacity (TAC) is in accordance with these findings. As a limitation of this study, measurements of MAO and DAO activities were performed in plasma and not in tissues, which may have implied a restricted assessment of the detoxification process of biogenic amines due to the consumption of experimental diets.

Hydrolyzed proteins may contain bioactive peptides with antioxidant activity, which may inhibit free radical scavenging, inactivate reactive oxygen species, and inhibit lipid peroxidation and metal ion chelation [29,30]. The consumption of HCLP diets did not alter the antioxidant activity in dogs, measured by TAC in which Cu^2+^ is reduced to Cu+. Previous studies have shown that the antioxidant activity of bioactive peptides is related primarily to the chelation of transition metals such as Fe^2+^ and Cu^2+^ [31]. Recently, a study demonstrated that the inclusion of an enzymatically hydrolyzed poultry by-product meal may reduce cat serum angiotensin-converting enzyme activity [32]. Future studies should emphasize the focus on bioactive peptides present in hydrolyzed proteins, due to their possible broad effects on the health of companion animals.

## 5. Conclusions

To our knowledge, this is the first study that provides the content of biogenic amines on the protein sources, diets, and their metabolism in dogs. The protein sources varied greatly in biogenic amine content. Based on our data, the poultry by-product meal had the highest content of putrefactive biogenic amines, such as putrescine, cadaverine, tyramine and histamine, leading to an increased consumption and balance of these compounds. In contrast, the content and intake of spermidine, phenylethylamine and spermine were higher for hydrolyzed chicken liver powder. Our results support previous evidence of the high efficiency of aminoxidases in the intestinal tract of dogs, given the lack of difference in plasma monoamine and diamine oxidase activities between the diets. Therefore, it appears that hydrolyzed chicken liver powder may be a desirable protein source in commercial foods for dogs with reduced ability to detoxify biogenic amines. Future studies should use a larger number of replicates and assess dietary biogenic amine content and plasma amino oxidase activities prior to starting the study.

## Figures and Tables

**Figure 1 animals-13-02578-f001:**
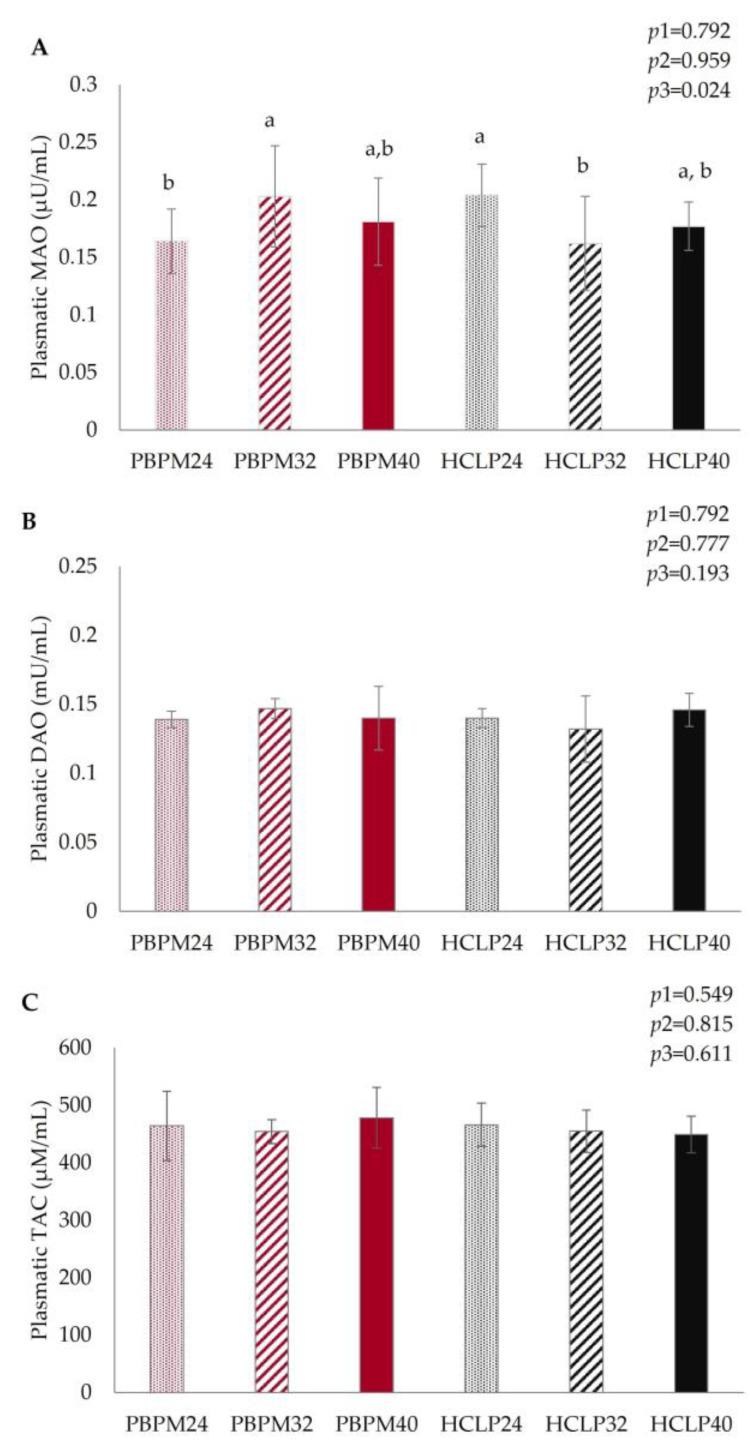
Plasmatic concentrations of (**A**) monoamine oxidase (MAO), (**B**) diamine oxidase (DAO) and (**C**) total antioxidant capacity (TAC) of dogs after 30 days of feeding with poultry by-product meal (PBPM, red columns) or hydrolyzed chicken liver (HCLP, black columns) diet at three crude protein concentrations (24, 32 or 40% CP). Results were expressed as mean ± standard deviation. Results were analyzed by two-way ANOVA. *p*1: comparison between protein sources (PBPM or HCLP); *p*2: comparison between different protein concentrations; *p*3: interaction between types of protein source and protein concentration. Different lowercase letters indicate significant differences according to Tukey post hoc test (*p* < 0.05 or 0.01).

**Table 1 animals-13-02578-t001:** Ingredient and chemical composition of experimental diets.

Ingredients, g/kg as Is	Diet ^1^					
	PBPM24	PBPM32	PBPM40	HCLP24	HCLP32	HCLP40
Brewers rice	54.9	44.9	34.6	52.5	42.2	30.5
Hydrolyzed chicken liver powder	-	-	-	14.4	21.9	31.0
Poultry by-product meal	12.5	20.3	28.7	-	-	-
Maize gluten meal	11.6	15.9	20.0	11.6	15.9	20.0
Swine blood plasma	1.95	2.66	3.38	1.95	2.66	3.38
Sugarcane fiber	3.2	3.11	3.00	3.65	3.80	4.00
Salt	0.50	0.50	0.50	0.50	0.50	0.50
Premix mineral/vitamin ^2^	0.40	0.40	0.40	0.40	0.40	0.40
Limestone	0.08	-	-	1.53	2.11	2.41
Dicalcium phosphate	2.05	1.14	0.08	1.56	0.65	0.16
Choline chloride	0.32	0.25	0.16	0.47	0.48	0.5
Potassium chloride	0.65	0.55	0.46	0.55	0.42	0.27
Phosphoric acid	0.50	0.50	0.50	0.50	0.50	0.50
Added by coating after extrusion, g/kg						
Poultry fat	9.87	8.33	6.69	8.85	7.02	4.98
Palatant enhancer ^3^	1.50	1.50	1.50	1.50	1.50	1.50
Analyzed chemical composition, % dry matter basis						
Dry matter	93.6	92.6	93.6	92.5	93.3	93.5
Crude protein	24.2	32.6	41.1	25.2	31.5	39.3
Acid-hydrolyzed fat	13.4	13.8	12.3	13.0	11.9	13.2
Ash	6.50	5.91	6.85	6.45	6.05	6.79
Crude fiber	10.2	7.83	9.04	5.80	9.88	6.15
Starch	47.7	37.9	31.0	44.5	39.2	30.4
Gelatinization index of starch	92.7	99.0	90.7	97.6	93.3	91.7
Gross energy, kcal/kg	4921	5164	5246	5023	4961	5166
Metabolizable energy, kcal/kg	3842	3464	3338	3537	3256	3586

^1^ PBPM24 = diet based on PBPM with 24% of protein; PBPM32 = diet based on PBPM with 32% of protein; PBPM40 = diet based on PBPM with 40% of protein; HCLP24 = diet based on HCLP with 24% of protein; HCLP32 = diet based on HCLP with 32% of protein; HCLP40 = diet based on HCLP with 40% of protein. ^2^ Premix mineral/vitamin (supplied per kilogram of diet): vitamin A (10,800 U), vitamin D3 (980 U), vitamin E (60 mg), vitamin K3 (4.8 mg), vitamin B1 (8.1 mg), vitamin B2 (6.0 mg), vitamin B6 (6.0 mg), vitamin B12 (30 μg), pantothenic acid (12 mg), niacin (60 mg), folic acid (0.8 mg), biotin (0.084 mg), manganese (7.5 mg), zinc (100 mg), iron (35 mg), copper (7.0 mg), cobalt (10 mg), iodine (1.5 mg), selenium (0.36 mg), choline (2.400 mg), taurine (100 mg), and antioxidant BHT (150 mg). ^3^ DTECH 8L, S.P.F. Argentina S.A., Argentina.

**Table 2 animals-13-02578-t002:** Biogenic amines concentration in ingredients and experimental diets analyzed and calculated (designated between parentheses based on the biogenic amines concentration of each ingredient).

	Biogenic Amines ^1^								Total Amines ^2^	BAI ^3^
	PUT	CAD	TYR	HIS	AGM	SPD	PHE	SPM		
Ingredient ^4^, mg/kg as is										
PBPM	648	784	556	39.3	33.7	34.3	ND	ND	2095	2027
HCLP	15.2	42.4	105	12.6	3.82	243	3.63	489	915	175
Diet ^5^, mg/kg as is										
PBPM24	69.8 (81.0)	87.3 (98.0)	68.2 (69.5)	9.73 (4.91)	ND (4.21)	5.78 (4.29)	5.87 (ND)	ND (ND)	247 (262)	235 (253)
PBPM32	132 (132)	152 (159)	119 (113)	9.70 (7.98)	7.09 (6.84)	13.6 (6.96)	3.46 (ND)	ND (ND)	437 (425)	413 (412)
PBPM40	174 (186)	218 (225)	161 (160)	18.4 (11.3)	9.80 (9.67)	13.5 (9.84)	ND (ND)	ND (ND)	595 (601)	572 (582)
HCLP24	ND (2.19)	0.40 (6.11)	0.43 (15.1)	ND (1.81)	ND (0.55)	ND (35.0)	ND (0.52)	0.67 (70.4)	1.50 (132)	1 (25)
HCLP32	12.5 (3.33)	18.5 (9.29)	36.5 (23.0)	5.97 (2.76)	1.39 (0.84)	48.2 (53.2)	5.99 (0.79)	53.4 (107)	183 (200)	73 (38)
HCLP40	15.0 (4.71)	17.4 (13.1)	39.1 (32.6)	6.14 (3.91)	1.71 (1.18)	53.0 (75.3)	6.82 (1.13)	75.2 (152)	214 (284)	78 (54)

ND = not detected (i.e., below detection limits). ^1^ PUT = putrescine; CAD = cadaverine; TYR = tyramine; HIS = histamine; AGM = agmatine; SPD = spermidine; PHE = phenylethylamine; SPM = spermine. ^2^ Total amines = putrescine + cadaverine + tyramine + histamine + serotonin + agmatine + spermidine + phenylethylamine + spermine + tryptamine. ^3^ BAI (biogenic amines index) = putrescine + cadaverine + tyramine + histamine and classified as: BAI < 5 mg/kg = good quality; 5 mg/kg < BAI < 20 mg/kg = acceptable; 20 mg/kg < BAI < 50 mg/kg = low quality; BAI > 50 mg/kg = spoiled. ^4^ PBPM = poultry by-product meal; HCLP = hydrolyzed chicken liver powder. ^5^ PBPM24 = diet based on PBPM with 24% of protein; PBPM32 = diet based on PBPM with 32% of protein; PBPM40 = diet based on PBPM with 40% of protein; HCLP24 = diet based on HCLP with 24% of protein; HCLP32 = diet based on HCLP with 32% of protein; HCLP40 = diet based on HCLP with 40% of protein.

**Table 3 animals-13-02578-t003:** Biogenic amines intake (mg/kg as is) of dogs fed a poultry by-product meal (PBPM) or a hydrolyzed chicken liver (HCLP) diet at three crude protein concentrations (24, 32 or 40% CP) ^1^.

		Biogenic Amines ^2^								Total Amines ^3^
		PUT	CAD	TYR	HIS	AGM	SPD	PHE	SPM	
PS ^4^	PBPM	153 ± 52.9 ^a^	186 ± 65.6 ^a^	142 ± 46.6 ^a^	15.4 ± 5.08 ^a^	6.82 ± 5.17 ^a^	13.4 ± 4.54 ^b^	3.88 ± 3.17 ^b^	ND ^b^	521 ± 175 ^a^
	HCLP	10.8 ± 8.06 ^b^	14.3 ± 10.2 ^b^	30.0 ± 21.7 ^b^	4.78 ± 3.51 ^b^	1.22 ± 0.91 ^b^	39.9 ± 29.4 ^a^	5.05 ± 3.74 ^a^	51.0 ± 38.4 ^a^	157 ± 115 ^b^
CPC ^5^	24%	44.1 ± 46.6 ^c^	55.4 ± 58.0 ^c^	43.3 ± 45.3 ^c^	6.14 ± 6.50 ^c^	ND ^c^	3.65 ± 3.86 ^b^	3.71 ± 3.92 ^b^	0.42 ± 0.45 ^c^	157 ± 164 ^c^
	32%	88.4 ± 77.0 ^b^	104 ± 86.0 ^b^	94.7 ± 53.9 ^b^	9.50 ± 2.58 ^b^	5.18 ± 3.69 ^b^	37.0 ± 21.6 ^a^	5.68 ± 1.59 ^a^	31.8 ± 33.4 ^b^	376 ± 168 ^b^
	40%	114 ± 101 ^a^	141 ± 127 ^a^	120 ± 77.8 ^a^	14.7 ± 7.92 ^a^	6.89 ± 5.15 ^a^	39.3 ± 24.5 ^a^	4.02 ± 4.22 ^b^	44.3 ± 46.6 ^a^	483 ± 246 ^a^
PBPM	24%	88.2 ± 10.7 ^c^	110 ± 13.4 ^c^	86.2 ± 10.5 ^c^	12.3 ± 1.50 ^b^	ND ^d^	7.30 ± 0.89 ^c^	7.41 ± 0.90 ^a^	ND ^c^	312 ± 38.0 ^c^
	32%	162 ± 7.48 ^b^	186 ± 8.61 ^b^	146 ± 6.75 ^b^	11.9 ± 0.55 ^b^	8.70 ± 0.40 ^b^	16.7 ± 0.77 ^b^	4.24 ± 0.20 ^b^	ND ^c^	536 ± 24.8 ^b^
	40%	209 ± 19.8 ^a^	262 ± 24.7 ^a^	193 ± 18.3 ^a^	22.1 ± 2.09 ^a^	11.8 ± 1.11 ^a^	16.1 ± 1.53 ^b^	ND ^c^	ND ^c^	714 ± 67.5 ^a^
HCLP	24%	ND ^e^	0.50 ± 0.06 ^d^	0.54 ± 0.07 ^e^	ND ^d^	ND ^d^	ND ^d^	ND ^c^	0.84 ± 0.10 ^c^	1.89 ± 0.23 ^e^
	32%	14.8 ± 1.49 ^de^	22.0 ± 2.21 ^d^	43.4 ± 4.35 ^d^	7.10 ± 0.71 ^c^	1.65 ± 0.17 ^c^	57.3 ± 5.75 ^a^	7.12 ± 0.71 ^a^	63.5 ± 6.37 ^b^	217 ± 21.8 ^d^
	40%	17.7 ± 1.53 ^d^	20.5 ± 1.78 ^d^	46.1 ± 3.99 ^d^	7.23 ± 0.63 ^c^	2.01 ± 0.17 ^c^	62.5 ± 5.41 ^a^	8.03 ± 0.70 ^a^	88.6 ± 7.68 ^a^	253 ± 21.9 ^cd^
*p*	PS	<0.0001	<0.0001	<0.0001	<0.0001	<0.0001	<0.0001	<0.0001	<0.0001	<0.0001
	CPC	<0.0001	<0.0001	<0.0001	<0.0001	<0.0001	<0.0001	<0.0001	<0.0001	<0.0001
	PS × CPC	<0.0001	<0.0001	<0.0001	<0.0001	<0.0001	<0.0001	<0.0001	<0.0001	<0.0001

ND = not detected (i.e., below detection limits). ^1^ Results were expressed as mean ± standard deviation. ^2^ PUT = putrescine; CAD = cadaverine; TYR = tyramine; HIS = histamine; AGM = agmatine; SPD = spermidine; PHE = phenylethylamine; SPM = spermine. ^3^ Total amines = putrescine + cadaverine + tyramine + histamine + serotonin + agmatine + spermidine + phenylethylamine + spermine + tryptamine. ^4^ PS (protein source): PBPM = poultry by-product meal; HCLP = hydrolyzed chicken liver powder. ^5^ CPC (crude protein concentration): 24, 32, and 40%. ^a–e^ Means in the same column with different lowercase letters are significantly different (*p* < 0.05).

**Table 4 animals-13-02578-t004:** Fecal excretion (mg/kg as is) of biogenic amines of dogs fed a poultry by-product meal (PBPM) or a hydrolyzed chicken liver (HCLP) diet at three crude protein concentrations (24, 32 or 40% CP) ^1^.

		Biogenic Amines ^2^								Total Amines ^3^
		PUT	CAD	TYR ^4^	HIS	SPD	PHE	SPM ^4^	TRY	
PS ^5^	PBPM	19.7 ± 17.5	3.92 ± 6.62	2.60 ± 4.08	0.40 ± 0.45	6.17 ± 4.43	0.36 ± 0.52	2.86 ± 6.12	0.59 ± 0.78	36.6 ± 28.0
	HCLP	21.7 ± 19.8	1.99 ± 4.03	1.68 ± 1.30	0.27 ± 0.34	10.7 ± 8.46	0.62 ± 0.46	2.70 ± 4.07	1.32 ± 1.26	41.0 ± 21.8
CPC ^6^	24%	16.3 ± 15.5	1.65 ± 2.42	1.65 ± 1.32	0.26 ± 0.31	10.9 ± 9.16	0.47 ± 0.49	0.97 ± 1.14	0.97 ± 1.14	34.4 ± 21.2
	32%	17.2 ± 17.5	2.00 ± 3.42	1.54 ± 1.83	0.33 ± 0.49	9.03 ± 6.71	0.53 ± 0.60	0.64 ± 0.98	0.64 ± 0.98	36.1 ± 22.3
	40%	28.7 ± 20.9	5.22 ± 8.35	3.24 ± 4.69	0.41 ± 0.41	5.45 ± 3.50	0.48 ± 0.44	1.25 ± 1.17	1.25 ± 1.17	46.1 ± 30.4
PBPM	24%	19.7 ± 14.1	1.90 ± 2.91	1.35 ± 0.63	0.24 ± 0.28	7.15 ± 4.85	0.63 ± 0.64	1.57 ± 3.85	0.25 ± 0.41	32.8 ± 19.7
	32%	16.1 ± 16.9	3.52 ± 4.41	2.17 ± 2.39	0.53 ± 0.62	6.95 ± 5.14	0.28 ± 0.50	6.02 ± 9.41	0.57 ± 0.89	36.1 ± 28.2
	40%	23.4 ± 23.1	6.33 ± 10.4	4.28 ± 6.71	0.45 ± 0.42	4.40 ± 3.33	0.18 ± 0.32	0.98 ± 2.41	0.96 ± 0.90	41.0 ± 38.0
HCLP	24%	12.8 ± 17.3	1.40 ± 2.07	1.94 ± 1.79	0.28 ± 0.36	14.6 ± 11.3	0.31 ± 0.21	2.85 ± 5.04	1.69 ± 1.21	35.9 ± 24.4
	32%	18.2 ± 19.6	0.48 ± 0.85	0.91 ± 0.81	0.14 ± 0.21	11.1 ± 7.89	0.78 ± 0.63	3.60 ± 4.4.60	0.72 ± 1.14	36.0 ± 17.4
	40%	34.1 ± 19.0	4.10 ± 6.46	2.19 ± 0.88	0.38 ± 0.43	6.49 ± 3.65	0.77 ± 0.32	1.64 ± 2.71	1.54 ± 1.41	51.2 ± 23.0
*p*	PS	0.719	0.719	0.639	0.308	0.052	0.113	0.377	0.050	0.583
	CPC	0.130	0.080	0.144	0.651	0.155	0.941	0.491	0.388	0.435
	PS × CPC	0.430	0.225	0.605	0.405	0.623	0.045	0.786	0.332	0.862

^1^ Results were expressed as mean ± standard deviation. ^2^ PUT = putrescine; CAD = cadaverine; TYR = tyramine; HIS = histamine; SPD = spermidine; PHE = phenylethylamine; SPM = spermine; TRY = tryptamine. ^3^ Total amines = putrescine + cadaverine + tyramine + histamine + serotonin + agmatine + spermidine + phenylethylamine + spermine + tryptamine. ^4^ Values transformed to ln for statistical analysis. ^5^ PS (protein source): PBPM = poultry by-product meal; HCLP = hydrolyzed chicken liver powder. ^6^ CPC (crude protein concentration): 24, 32, and 40%.

**Table 5 animals-13-02578-t005:** Balance of biogenic amines of dogs fed a poultry by-product meal (PBPM) or a hydrolyzed chicken liver (HCLP) diet at three crude protein concentrations (24, 32 or 40% CP) ^1^.

		Biogenic Amines ^2^									Total Amines ^3^
		PUT	CAD	TYR	HIS	AGM	SPD	PHE	SPM	TRY	
PS ^4^	PBPM	133 ± 53.9 ^a^	182 ± 63.7 ^a^	139 ± 45.2 ^a^	15.0 ± 5.04 ^a^	6.82 ± 5.17 ^a^	7.20 ± 6.70 ^b^	3.52 ± 3.01 ^b^	−2.86 ± 6.12 ^b^	−0.59 ± 0.78	484 ± 172 ^a^
	HCLP	−10.9 ± 18.5 ^b^	13.4 ± 10.3 ^b^	30.3 ± 20.9 ^b^	4.51 ± 3.55 ^b^	1.22 ± 0.91 ^b^	29.2 ± 33.0 ^a^	4.43 ± 3.53 ^a^	48.3 ± 38.9 ^a^	−1.32 ± 1.26	116 ± 113 ^b^
CPC ^5^	24%	27.8 ± 44.6 ^b^	59.1 ± 57.4 ^c^	46.0 ± 45.2 ^c^	5.88 ± 6.51 ^c^	0 ^c^	−7.24 ± 11.3 ^b^	3.24 ± 3.77 ^b^	−1.79 ± 4.27 ^c^	−0.97 ± 1.14	122 ± 165 ^c^
	32%	71.2 ± 80.1 ^a^	102 ± 84.6 ^b^	93.2 ± 53.3 ^b^	9.17 ± 2.44 ^b^	5.18 ± 3.69 ^b^	28.0 ± 19.5 ^a^	5.15 ± 1.43 ^a^	26.9 ± 35.3 ^b^	−0.64 ± 0.98	340 ± 169 ^b^
	40%	84.8 ± 108 ^a^	136 ± 126 ^a^	116 ± 76.5 ^a^	14.3 ± 7.88 ^a^	6.89 ± 5.15 ^a^	33.9 ± 23.7 ^a^	3.54 ± 3.92 ^b^	43.0 ± 46.4 ^a^	−1.25 ± 1.17	437 ± 252 ^a^
PBPM	24%	68.5 ± 10.3 ^c^	108 ± 13.5 ^c^	84.8 ± 10.2 ^c^	12.1 ± 1.37 ^b^	0 ^d^	0.15 ± 4.64 ^c^	6.78 ± 1.04 ^a^	−1.57 ± 3.85 ^c^	−0.25 ± 0.41	279 ± 28.8 ^c^
	32%	146 ± 18.5 ^b^	183 ± 12.1 ^b^	144 ± 7.88 ^b^	11.4 ± 0.92 ^b^	8.70 ± 0.40 ^b^	9.72 ± 4.89 ^bc^	3.96 ± 0.41 ^b^	−6.02 ± 9.41 ^c^	−0.57 ± 0.89	400 ± 33.9 ^b^
	40%	186 ± 29.8 ^a^	255 ± 22.1 ^a^	189 ± 15.8 ^a^	21.7 ± 1.91 ^a^	11.8 ± 1.11 ^a^	11.8 ± 3.87 ^b^	−0.18 ± 0.32 ^c^	−0.98 ± 2.41 ^c^	−0.96 ± 0.90	673 ± 68.8 ^a^
HCLP	24%	−12.8 ± 17.3 ^d^	−0.07 ± 0.53 ^d^	−0.67 ± 0.30 ^e^	−0.28 ± 0.36 ^d^	0 ^d^	−14.6 ± 11.3 ^d^	−0.31 ± 0.21 ^c^	−2.01 ± 5.02 ^c^	−1.69 ± 1.21	34.0 ± 24.3 ^e^
	32%	−3.42 ± 20.4 ^d^	21.5 ± 2.43 ^d^	42.5 ± 4.77 ^d^	6.96 ± 0.74 ^c^	1.65 ± 0.17 ^c^	46.2 ± 3.56 ^a^	6.34 ± 0.96 ^a^	59.9 ± 6.51 ^b^	−0.72 ± 1.14	181 ± 29.0 ^d^
	40%	−16.4 ± 18.6 ^d^	16.4 ± 7.66 ^d^	43.9 ± 4.63 ^d^	6.86 ± 0.86 ^c^	2.01 ± 0.17 ^c^	56.0 ± 7.13 ^a^	7.26 ± 0.68 ^a^	87.0 ± 8.89 ^a^	−1.54 ± 1.41	201 ± 32.4 ^d^
*p*	PS	<0.0001	<0.0001	<0.0001	<0.0001	<0.0001	<0.0001	<0.0001	<0.0001	0.050	<0.0001
	CPC	<0.0001	<0.0001	<0.0001	<0.0001	<0.0001	<0.0001	<0.0001	<0.0001	0.388	<0.0001
	PS × CPC	<0.0001	<0.0001	<0.0001	<0.0001	<0.0001	<0.0001	<0.0001	<0.0001	0.332	<0.0001

^1^ Results were expressed as mean ± standard deviation. ^2^ PUT = putrescine; CAD = cadaverine; TYR = tyramine; HIS = histamine; AGM = agmatine; SPD = spermidine; PHE = phenylethylamine; SPM = spermine; TRY = tryptamine. ^3^ Total amines = putrescine + cadaverine + tyramine + histamine + serotonin + agmatine + spermidine + phenylethylamine + spermine + tryptamine. ^4^ PS (protein source): PBPM = poultry by-product meal; HCLP = hydrolyzed chicken liver powder. ^5^ CPC (crude protein concentration): 24, 32, and 40%. ^a–e^ Means in the same column with different lowercase letters are significantly different (*p* < 0.05).

## Data Availability

Not applicable.

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
