# Peer review of "Effects of Diets Based on Hydrolyzed Chicken Liver and Different Protein Concentrations on the Formation and Deamination of Biogenic Amines and Total Antioxidant Capacity of Dogs"

_animals, 2023, doi:10.3390/ani13162578_

Round 1

Reviewer 1 Report

The manuscript present results obtained after feeding dogs with 2 types of dog diets (i.e. 2 protein sources, one chicken liver based, one poultry carcass based)  with 3 different concentration of the protein sources; i.e. a total of 6 different formulations. 12 animals are included in the study, in three turns (n=2 group per turn; so that ultimately 6 individuals receive same feeding) with 30 days feeding each. Before the feeding trials, a 30 days of conventional feeding precede.

It would have been interesting (although requiring a different experimental/statistical setup) to have the BA feces and aminooxidase values before experimental feeding as a sort of baseline. 

The calculated as well as analysed composition and amine contents of the feeds are displayed in tables. The amine contents excreted in the feces are displayed. The changes in blood plasma amine oxidase as well as total antioxidant activity are measured. Ingested amines and excreted amines are compared to assess, with a negative balance (i.e. less excreted than ingested) indicative for deamination (and absorption?) and a positive balance indicating increased lumilnal amine formation.

Data are analysed by ANOVA; factors and interactions are well defined. Post-hoc test is described.

There are some issues that need to be addressed:

line 20: .. and may cause other toxicological effects..

lines 46-47 and others: in the text body, the authors describe the Mietz&Karmas definition, but the limits they present (e.g. 50 mg/kg..) are Ruiz-Capillas or similar authors. Mietz&Karmas present a dimensionless index, whereas other authors use sums, in mg/kg - see also Table 2, footnote 5.

lines 64-65: The use of BAI etc. is NOT because of toxicological effects! Histamine limits are examples for toxicology-based limits.

line 67: "handling conditions" : it is "time and temperature"

line 76: is it realistic that hydrolysed proteins will be handled in a way that they can "be contaminated" with biogenic amines? Isn´t it more likely that the raw material before processing will spoil?

line 121: 10,800 U (blank space before "U")

line 123:  vitamin B12?; "(30 µg)"

line 125: 2,400 mg

line 135: 10,000 g

line 139: HPLC analysis details should be given. Derivatization? RP-HPLC, detection? Internal standard? Which amines tested (this can now only be inferred from table footnotes)

line 172:  "highest concentrations"

line 172: it should be defined what putrefactive amines are (now this definition is  footnote 3 to Table 2; please give a reference for that; also, why eg tyramine is not indicating putrefaction/spoilage)

Table 2, footnote 4: spermine, spermidine can be seen as freshness indicators, but the organ-specific concentrations may differ substantially. So, liver has higher polyamine content than e.g. muscle, even when it would be less "fresh", for example when stored for a longer period of time.

Table 2, footnote 5: the Mietz&Karmas definition given here does not match with the limits applied.

Table 3, header: mg/kg means mg/kg body mass?

Table 4, header: mg/kg means mg/kg feces?

line 278, Figure 1: "post-hoc"

lines 280-282: "quality index" does not (only) refer to toxicological effects, please rephrase.

line 305: remove "in".

When you feed liver based protein sources compared to muscle based ones (or partially deboned poultry carcasses), the higher polyamine content in liver based diets can be expected and is not surprising ((poly)amine contents e.g. in works from P. Kalac)

lines 310-312: the other amines do not exclusively originate from decomposition processes, there are some more or less small, but detectable quantities of these other amines, so maybe better "while the other amines can also be formed by protein degradation driven by bacteria."

line 395: the family name must read: Huis in´t Veld The correct authors list is:

Ten Brink, B.; Damink, C.; Joosten, H.M.L.J.; Huis in´t Veld, J.H.J.

Reviewer 2 Report

I read with pleasure your manuscript “Effects of diets based on hydrolyzed chicken liver and different protein concentrations on the formation and deamination of biogenic amines and total antioxidant capacity of dogs”. Overall, the is manuscript is well prepared and concise and includes all the related info. This work is relevant to this journal. The following comments will help improve the manuscript.

Dear Authors, some specific comments:

-        in keywords add: dog diets, poultry by-product meal, hydrolyzed chicken liver powder and delate: high protein diets; hydrolyzed protein

-        add hypothesis at the beginning

-        in table 1 – units: Dry matter is in % dry matter basis? correct, in title delate units ‘g/kg’, add more information about ME calculated and about swine blood plasma (source and more specifications – e.g. spray-dried)  in M&M

-        add reference to Statistical analysis

-        Discussion: overall, this manuscript needs more discussion about experimental results

-        Conclusions:  Need to be rewritten. It is too fast and does not include the positive and negative elements highlighted in your manuscript. Make sure the conclusion is short and solid. An idea may be to synthetize in 3-5 bullet the key results of the study, evidences and recommendation. This improvement will increase clearness and readability. Add a practical implications statement

Reviewer 3 Report

Effects of diets based on hydrolyzed chicken liver and different 2 protein concentrations on the formation and deamination of bi- 3 ogenic amines and total antioxidant capacity of dogs

The paper is well written and provides information on the effect of using different proteic ingredients in dog diets. The number of repetitions in small which is one of the limitations of the study, but authors analyses biogenic amines at different levels, from the diet to the faeces with consistent results. Authors should provide further information in the material and methods section. Some other aspects have to be taken into consideration to improve the paper quality. See the comment below:

Line 103-115. Include information on the provider for the HCLP and PBPM and quality control of these ingredients. Where was the feed made?(Include information of the Company)

Line  138 Biogenic amine contents were determined in the supernatant by 138 HPLC (HPLC model LC-10AD; Shimadzu Corporation, Kyoto, Japan).

Please, Include the HPLC conditions and HPCL characteristics for the analysis (column, solvents, gradients etc)

Line 201. Dogs fed PBPM had higher concentrations of putrescine, cadaverine, tyramine, his- 201 tamine, agmatine, and the sum of total amines HCLP-based diets (p < 0.0001) (Table 3).

Concentration in plasma, serum, faeces??. Clarify

TAble 3. It is not clear if these biogenic amines concentration is in blood??. It is said “intake” and then what is the different with Table 2 that present data of ingredients and diets?. Clarify.

Line 213. Lastly, dogs fed with HCLP at 32 and 213 40% CP had greater fecal excretion of phenylethylamine (p = 0.045).

This is an interaction effect. Explain better. Tukey letters should appear in TAble 4.

Line 250. Figure 1D is missing

Lines 249-253. You should explain interactions or give more detailed information as observed in Table.

Lines 306-307. “only con- 306 sumed more spermidine, phenylethylamine, and spermine than dogs fed PBPM-based 307 diets”

It is not clear how did you measured the consumption and what these data represent (level in blood?). You should clarify all this information in the paper (material and methods, results and discusion).

Reviewer 4 Report

Paper is well written and easy to follow. The results are pretty straightforward.

I have a hard time distinguishing between Table 2 and Table 3 - aren’t they both reporting the measures in the diets? Shouldn’t intakes and diets expressed as mg/kg as is be identical?

Table 3 - the letter c in SPM at CPC 24% should be a superscript.
